# Effect of Ciprofloxacin on the Composition of Intestinal Microbiota in *Sarcophaga peregrina* (Diptera: Sarcophagidae)

**DOI:** 10.3390/microorganisms11122867

**Published:** 2023-11-27

**Authors:** Haojie Tang, Xiangyan Zhang, Fengqin Yang, Changquan Zhang, Fernand Jocelin Ngando, Lipin Ren, Yadong Guo

**Affiliations:** Department of Forensic Science, School of Basic Medical Sciences, Central South University, Changsha 410013, China; 226511097@csu.edu.cn (H.T.); zxy196@csu.edu.cn (X.Z.); yangfengqin@csu.edu.cn (F.Y.); zcq120@csu.edu.cn (C.Z.); 206508005@csu.edu.cn (F.J.N.)

**Keywords:** *Sarcophaga peregrina*, ciprofloxacin, developmental cycle, gut microbiota

## Abstract

The intestinal bacteria of insects are crucial to the growth and development of the host. It has been found that various physiological processes of insects, such as immune response, metabolism, reproductive ability, and growth and development, involve the gastrointestinal flora. However, many external factors affect the composition of insects’ intestinal microorganisms, such as the type of dietary substrate. *Sarcophaga peregrina* (Robineau-Desvoidy, 1830) (Diptera: Sarcophagidae) is of great significance in medicine and forensic science. In this study, we investigated the effects of ciprofloxacin on the growth and gut microbiota of *S. peregrina*. The results demonstrated that the maximum body length of larvae was not affected by ciprofloxacin, while the growth rate of body length quickened as the concentration of the drug increased. The weight of the pupa and adult was reduced significantly due to the effect of ciprofloxacin. After analyzing the gut microbiota composition of *S. peregrina* in different drug groups, it was indicated that *Ignatzschineria*, *Providencia*, *Wohlfahrtiimonas*, *Proteus*, *Myroides*, and *Bacteroides* play important roles in the growth of *S. peregrina*. However, they still need to be further studied. In general, ciprofloxacin can affect the gut microbial community structure, which in turn affects the fitness of the host.

## 1. Introduction

Microorganisms are crucial for the physiological functions of an insect’s immune system [1], metabolism [2], and reproductive capacity [3,4]. Therefore, the development of insects is closely related to the biological processes of intestinal microbes. Existing studies have found that in rearing *Drosophila* in either germy or germ-free environments, the commensal microbiota is able to influence their systemic development by affecting both growth rate and body size [5]. *Acetobacter* and *Lactobacillus* can promote the development of *Drosophila* larvae, especially when nutrients are deficient [6,7]. *Burkholderia* symbiont promotes the development of *Riptortus pedestris* [4]. However, many extrinsic factors, such as the type of food substrate [7] and antibiotics, seriously impact the composition of gut microbes, which in turn affects growth and development [8,9].The habit of foraging and reproducing on corpses, decaying organic matter, and even feces lead necrophilic insects to have complex and diverse symbiotic microorganisms. The symbiotic microbiome of *Lucilia sericata* [10,11,12], *Lucilia cuprina* [10,13], *Chrysomya megacephala* [14], *Musca domestica* [14,15], and *Phormia regina* [16] has been investigated. With the escalating misuse of antibiotics, the toxicological impacts of these drugs demand sufficient consideration. Regrettably, research on the effects of antibiotics on insect commensal flora and development remains scarce.

Ciprofloxacin, a member of the esteemed quinolone antibiotic family, proudly stands as the fifth most prolific generic antibiotic in the world [17,18]. Its versatile applications encompass the treatment of respiratory, urinary, gastrointestinal, and abdominal infections caused by an array of both Gram-positive and Gram-negative bacteria [19,20,21,22,23]. Fascinating studies on insect growth and development have uncovered that ciprofloxacin possesses the ability to neutralize *Bacillus sphaericus*, thereby fostering the proliferation of *Aedes* (*Stegomyia*) *aegypti* larvae [24]. Similarly, in the realm of *Chrysomya pretoria*, Ferraz et al. [25] unearthed a stimulatory effect of ciprofloxacin on growth. Conversely, the metamorphosis of *Anopheles barbirostris* (Diptera: Culicidae) from larvae to adult is notably hindered by the administration of ciprofloxacin [26]. In addition, investigations into *Calliphora vomitoria* revealed that levofloxacin impeded maggot development, thereby shedding light on the intricate relationship between antibiotics and the intricate process of insect maturation [27]. Concerning *Chrysomya megacephala*, treatment with gentamicin did not yield significant alterations in its developmental period [28]. These remarkable findings imply a potential interplay between antibiotics and the host gut microbiota, influencing insect growth and development. Hence, it becomes imperative to investigate the effects of antibiotics on the growth and development of flies.

In this study, we investigate *Sarcophaga peregrina* (Robineau-Desvoidy, 1830), a member of the Sarcophagidae family (known as flesh flies). These are closely associated with human life in ecological habitats [29], and widely spread from tropical to subtropical areas of the Palaearctic, Oriental, and Oceanian regions [30]. Adults of *S. peregrina* were procured from their natural habitat, and subsequently, a laboratory population was established. Essential developmental information on *S. peregrina* was gathered under a constant temperature (25.0 °C), with and without the inclusion of ciprofloxacin in their diet. Subsequently, the sequencing of gut symbiotic microbiota was conducted by targeting the V3–V4 variable region of the 16S rRNA gene, encompassing the gut microbiota of *S. peregrina* in five distinct stages. In a broader sense, the impact of varying concentrations of ciprofloxacin on the growth of *S. peregrina* was thoroughly investigated. Furthermore, the composition of *S. peregrina*’s gut microbiome was elucidated, with an exploration of the potential correlation between changes in gut microbiota and macroscopic growth. The findings of this study will serve as a valuable point of reference for understanding the influence of microorganisms on host growth and development in future investigations.

## 2. Materials and Methods

### 2.1. Experimental Design

The incubation period was considered the starting point of the developmental phase of *S. peregrina*. Upon hatching, the newborn larvae were meticulously divided into three groups: the control group (Z), experimental group 1 (H1), and experimental group 2 (H2). A diet of minced meat, composed of fresh pig lung weighing approximately 600 g, was presented as their daily sustenance. The concentrations of ciprofloxacin in each group were meticulously measured: Z = 0 μg/g, H1 = 0.111 μg/g, and H2 = 1.33 μg/g, respectively. The sampled larvae were meticulously scrutinized to determine their instar, relying upon the number of clefts in the posterior spiracle. The larval stage encompassed the first, second, and third instar. Once the third instar was reached, the larvae embarked on their wandering stage. It is during this phase that the wandering larvae inevitably undergo metamorphosis, a mesmerizing transformation known as the pupal period, ultimately emerging as adults. Samples were diligently gathered at fixed intervals throughout the day (8:00 a.m., 3:00 p.m., and 11:00 p.m.). A trilogy of replicates was performed, ensuring the integrity of the results.

The collection of intestinal samples of *S. peregrina* was conducted in a pristine and sterile environment. Initially, the surface of the organism’s body was meticulously cleansed with a solution containing 75% alcohol and 0.05% sodium hypochlorite, allowing for a thorough disinfection period lasting one minute. Subsequently, the cleansed surface was rinsed three times with a sterilized solution of PBS (pH 7.4), followed by an additional three to five rinses with sterile water. The dissection of both larvae and adults was meticulously performed within the confines of the PBS solution.

### 2.2. Cultivation of S. peregrina

The adult *S. peregrina* specimens were trapped in Changsha, Hunan Province, China, lured by pork lung bait. This served as a fertile medium for their ovipositing and larval rearing. Following two successive generations, specific purified species were meticulously chosen for subsequent experiments. Within each generation, adult *S. peregrina* pairs were housed in an artificial climate chamber, where the temperature was carefully maintained at 25 ± 1 °C, accompanied by a relative humidity of 70 ± 10%. The gentle rhythm of 12:12 h light and darkness created the perfect ambiance.

### 2.3. 16S rRNA Microbial Community Analysis

A total of 45 samples were gathered, comprising third-instar larvae (n = 9, Z-L3, H1-L3 H2-L3); prepupal stage (n = 9, Z-PR, H1-PR, H2-PR); early pupal stage (n = 9, Z-PE, H1-PE, H2-PE); late pupal stage (n = 9, Z-PL, H1-PL, H2-PL); and recently emerged adults (n = 9, Z-A, H1-A, H2-A). All DNA was extracted from the combined digestive systems utilizing an E.Z.N.A. Soil DNA KF Kit (Omega Bio-tek, Norcross, GA, USA) following the manufacturer’s guidelines. PCR amplification of the V3–V4 region of the 16S rRNA gene was chosen using universal bacterial primers (338F: 5′-ACTCCTACGGGAGGCAGCAG-3′; 806R: 5′- GGACTACHVGGGTWTCTAAT-3′). The PCR products were purified using an AxyPrep DNA Gel Extraction Kit (Axygen Biosciences, Union City, CA, USA) and quantified with a Quantus™ Fluorometer (Promega, Madison, WI, USA). Sequencing was conducted on an Illumina Miseq PE300/NovaSeq PE250 (Meiji Biomedical Technology Co., Ltd. Shanghai, China).

### 2.4. Quality Control of Sequencing Data

The raw image data files acquired via high-throughput sequencing underwent base identification (base-calling), resulting in raw sequencing sequences. These sequences, referred to as raw data, were then subjected to initial scanning using the sliding window method (version 0.35) [31]. In this process, windows with an average base mass below the threshold of 20 were eliminated, along with sequences shorter than 50 base pairs. Subsequently, the qualified paired-end raw data from the previous step were merged using Flash (version 1.2.11) [32] software, with a maximum overlap of 200 base pairs, yielding complete paired-end sequences. To obtain clean tag sequences, split_libraries (version 1.8.0) [33] software within QIIME was employed. This software removed sequences in the paired-end sequences that contain N-bases, and sequences with single base repeats exceeding 8 and lengths below 200 base pairs. UCHIME (version 2.4.2) [34] software was then utilized to eliminate any chimeric tags present in the clean tags. Finally, valid tags suitable for subsequent OTU partitioning were obtained. OE Biotech Co., Ltd. (Shanghai, China) conducted the 16S rRNA sequencing and analysis.

### 2.5. Taxonomic Classification of OTUs

Search software (version 2.4.2) [35] was utilized to categorize the OTU valid tags obtained from quality control sequences based on a 97% similarity threshold. Subsequently, the sequences with the highest abundance within each OTU were chosen as the representative sequences of the OTU. Next, an RDP classifier employed the naive Bayesian classification algorithm [36] to compare the annotation of the representative sequence with the Silva (V138) database, yielding the annotation information for the OTU. Finally, the abundance matrix file for each OTU in every sample was constructed by considering the number of sequences contained within each OTU across the samples.

### 2.6. Statistical Analysis

The data collected during the experiment were processed and analyzed using SPSS software(R26.0.0.0), with a predetermined significance level of *p* = 0.05 for all statistical evaluations. To identify differences in baseline data development across the three groups, a one-way analysis of variance was conducted. We utilized the Specaccum species accumulation curve to assess whether the sample size was adequate. The Simpson index was then used to characterize the diversity and evenness of species distribution within the community. Additionally, Good’s coverage index served as an indicator of sequencing depth, with values approaching 1 signifying a more reasonable sequencing depth. A principal coordinates analysis (PcoA) and non-metric multidimensional scaling analysis (NMDS) were applied to compare differences among different sample groups. Finally, Adonis (PERMANOVA) and ANOVA analyses were utilised to ascertain significant differences among the groups and identify the species exhibiting such differences.

## 3. Results

### 3.1. Effects of Ciprofloxacin on Body Length and Body Weight of S. peregrina

Certain trends began to emerge in terms of larval body length, although no statistically significant distinctions were observed among the groups (Figure 1 and Table 1). The H2 group exhibited the most pronounced growth rate, followed by the H1 group, while the Z group displayed the smallest growth rate (between 18 and 56 h). However, there were no significant differences in terms of the maximum larval body length (*p* > 0.05) among the three groups (Table 1). The maximum body lengths were as follows: Z group—19.74 ± 0.33 mm, H1 group—19.60 ± 0.91 mm, and H2 group—19.11 ± 0.56 mm, respectively.

Pupae and adults from the Z, H1, and H2 groups were assembled and assessed in terms of their body mass. A remarkable divergence emerged in the weight of pupae when contrasting groups Z and H1 (with Z surpassing H1, *p* < 0.05), and when comparing groups Z and H2 (with Z surpassing H2, *p* < 0.05) (Table 1). Conversely, no substantial variation in weight was detected between group H1 and group H2 (*p* > 0.05). Concerning the body mass of the mature specimens, the minimum value was documented within the H2 group, which exhibited a significant difference from the other two groups (*p* < 0.05). However, no noteworthy disparities were discerned between groups Z and H1 (*p* > 0.05).

### 3.2. Gut Microbiota Sequencing Results and Data Analysis

#### 3.2.1. Sequencing Results

Briefly, 16S rRNA gene amplicon sequencing was conducted to acquire the gut microbial composition of *S. peregrina* within various cohorts. After thorough quality filtering and the elimination of chimeric sequences, a dataset consisting of 65,074 to 72,387 valid sequences was obtained, each with an average sequence length ranging from 416.51 to 426.73 base pairs (Appendix A). The observed number of operational taxonomic units (OTUs) varied from 229 to 740 in each sample, resulting in a total of 2559 OTUs. Further ANOVA analysis revealed 169 differential OTUs, with 9 and 96 of them occurring at the phylum and genus levels, respectively (Appendix A). Additionally, a total of 153 OTUs were found to be shared among different experimental groups (Figure 2).

#### 3.2.2. Gut microbiota Composition

All operational taxonomic units (OTUs) have been identified to encompass an impressive array of 26 phyla, 59 classes, 146 orders, 242 families, 461 genera, and 694 species (Appendix A). To unveil the relative abundance of the preeminent 15 bacteria at both the phylum and genus levels, rigorous analyses were conducted (Appendix A and Figure 3). At the phylum level, the consortia of commensal microbiota predominently comprised *Proteobacteria*, *Bacteroidetes*, *Firmicutes*, *Actinobacteria*, *Fusobacteriota*, *Desulfobacterota*, *Campilobacterota*, *Gemmatimonadota*, *Myxococcota*, *Acidobacteriota*, *Spirochaetota*, *Deferribacterota*, *Fibrobacterota*, and *Bdellovibrionota* (Figure 3a). Remarkably, the genera that reigned supreme were *Ignatzschineria*, *Providencia*, *Myroides*, *Proteus*, *Bacteroides*, *Wohlfahrtiimonas*, *Muribaculaceae*, *Escherichia-Shigella*, *Halomonas*, *Prevotella*, *Koukoulia*, *Lachnospiraceae NK4A136 group*, *Bifidobacterium*, *Ralstonia*, and *Lactobacillus* (Figure 3b).

Alpha diversity analysis (Figure 4) serves as a foundation for juxtaposing the magnitudes of diversity among samples. The Specaccum species, portrayed through its curve (Figure 4a), tends towards a plateau, implying that the augmentation of species within this milieu would not witness a substantial surge alongside the expansion of the sample size, thus affirming the sufficiency of the sampling effort. The curve of the Shannon index eventually reaches a steady state, a testament to the ample sequencing depth attained within the sample (Figure 4b). The Good’s coverage index, akin to the extent of unity (Figure 4c), signifies the comprehensive scope of sequencing, encapsulating all the species within the sample.

Beta diversity is employed to examine dissimilarities among samples (Figure 5). Initially, PcoA and NMDS (Figure 5) were performed to assess distinctive sequences. Bray–Curtis distances were utilized to construct a phylogenetic distance matrix, and the outcomes were visualized based on two-dimensional principles. Clustering of samples on NMDS plots signifies the presence of similar microbial community compositions. Subsequently, an Adonis analysis substantiated significant variations among the groups (*p* < 0.05, Table 2). To ascertain the dissimilarities in microbial composition across various levels between the groups, we employed an ANOVA analysis. A boxplot analysis illustrated the juxtaposition of the foremost 10 prevailing bacterial species, at both the phylum and genus levels, in relation to their proportional prevalence amongst the cohorts. (Appendix A, Appendix A). Line charts, designed based on the disparities in individual genera, aptly illustrated the divergence (Figure 6). Regarding *Ignatzschineria*, the overall relative abundance diminished as the developmental process unfolded. However, it exhibited an upsurge in L3 when subjected to ciprofloxacin, only to dwindle to nearly negligible levels after pupation (Figure 6a). *Myroides* exhibited heightened relative abundance during the PR, PE, and PL stages. The overarching pattern across these three groups was similar: relative abundance increased from L3 to PL, yet experienced a significant decline in the adult stage (Figure 6b). *Bacteroides* saw a progressive increase in relative abundance from PR onwards, maintaining an upward trajectory across all groups (Figure 6c). *Proteus* predominantly manifested during the PR stage and beyond (Figure 6d). *Providencia*’s relative abundance reached its nadir during the PR stage in all three groups, with irregular shifts observed in subsequent stages (Figure 6e). *Wohlfahrtiimonas* exhibited prevalence in L3 and PR, yet its relative abundance diminished as the concentration of ciprofloxacin treatment increased (Figure 6f). The attached file shows the remaining four microorganisms at the genus level, namely *Halomonas*, *Escherichia−Shigella*, *Muribaculaceae*, and *Prevotella* (Appendix A). Their low relative abundance and lack of significant variation patterns justify their inclusion in this presentation.

## 4. Discussion

In this investigation, the presence of ciprofloxacin in food substrates showed the potential to impact the growth and development of *S. peregrina*. Pupal weight exhibited a significant decrease under the influence of ciprofloxacin treatment, while no noteworthy disparity was observed among the various treatment concentrations. Similar outcomes were obtained when *Lucilia sericata* were subjected to different levels of levofloxacin [37]. It is conceivable that ciprofloxacin exerts an influence on growth and development.

Concerning adult weight, H1 displayed a lower value compared to that of group Z, albeit lacking statistical significance. However, the weight of H2 was significantly inferior to that of groups Z and H1. Similarly to the preceding investigation, the growth of *L. sericata* larvae remained unaffected by seven antibiotics, namely mezlocillin ampicillin, vancomycin, ceftizoxime, clindamycin, gentamicin, and cefazolin, when administered at therapeutic doses [38]. Our findings suggest that the development of adults is not significantly impeded unless the concentration of ciprofloxacin treatment falls within the range of 0.111 μg/g to 1.33 μg/g. While the maximum body length of larvae remains unaffected by ciprofloxacin, the rate of body length growth accelerates during the period between 18 and 56 h as the drug concentration increases. Further investigation is warranted to elucidate the specific underlying reasons for this.

At the phylum level, the dominant microbial groups inhabiting the gastrointestinal tract of *S. peregrina* are *Proteobacteria*, *Bacteroidota*, and *Firmicutes*. At the genus level, *Ignatzschineria*, *Providencia*, *Myroides*, and *Proteus* prevail, aligning with the findings of Gupta et al. [39]. Past investigations have proposed that gut symbionts establish stable colonization within specific niches, exhibiting variations across different fly species. Pais and colleagues have demonstrated the consistent presence of *Acetobacter thailandicus* in the intestinal tract of *D. melanogaster* [40]. *Enterobacteriaceae*, particularly *Klebsiella* and *Enterobacter*, dominate the gut bacterial community in various fruit flies, including *A. serpentine*, *A. obliqua*, *A. strata* [41], and *Bactrocera oleae* [42]. The outcomes of this experiment reveal a relatively higher diversity in the gut microbiome composition of *S. peregrina* compared to previously reported Diptera species. This could be attributed to the necrophilic nature of these flies, which predisposes them to associate with decaying matter. Furthermore, dietary and environmental factors may influence the diversity of the gut microbiome in these host insects [43,44]. The bacterial diversity in *S. peregrina* varies throughout different life stages, encompassing eggs, larvae, pupae, and adults, mirroring the patterns observed in other fully metamorphosed dipteran insects, such as *Parasarcophaga similis*, *L. sericata*, *Pholia regina* [45], and *Musca domestica* [44].

The microbial composition exhibited variations among distinct cohorts at each developmental phase, resembling prior investigations, thereby implying the potential existence of distinct roles for diverse bacterial species. *Ignatzschineria*, *Providencia*, and *Wohlfahrtiimonas* thrive abundantly during the L3 stage of *S. peregrina*, suggesting their association with this particular stage. However, the effects of ciprofloxacin on these bacteria diverge. As the concentration of ciprofloxacin increases, the relative abundance of *Ignatzschineria* also increases, while *Wohlfahrtiimonas* decreases. The behavior of *Providencia* exhibits a twofold variation, decreasing at low doses and increasing at high doses. During the PR period, the primary bacteria associated are *Ignatzschineria*, *Wohlfahrtiimonas*, and *Providencia*. However, only *Providencia* experiences a significant decrease during this stage, hinting at potential harm to the wandering stage of *S. peregrina*. It is noteworthy that common necrophilic fly larvae, such as *Chrysomya megacephala* and *M. domestica,* which feed on wheat bran [46], also harbor *Ignatzschineria* as the dominant genus in their gut microbiome. This implies that *Ignatzschineria* may play a pivotal role in fly development. Additionally, we hypothesize that *Proteus*, *Myroides*, *Providencia*, and *Bacteroides* are primarily associated with the pupal stage, with *Proteus* being more abundant during the prepupal stage and *Myroides* being more abundant during the late pupal stage. *Providencia* and *Bacteroides* emerge as the main bacteria related to the adult stage of *S. peregrina*. However, their relative abundance fluctuates with the concentration of ciprofloxacin, whereby the relative abundance of *Providencia* decreases as ciprofloxacin concentration increases, while *Bacteroides* exhibits the opposite trend.

## 5. Conclusions

The present investigation aimed to assess the impact of ciprofloxacin on the growth and composition of gut microorganisms during various development stages of *S. peregrina*. It was observed that the maximum body length of larvae remained unaffected by ciprofloxacin, while the growth rate of body length accelerated with increasing drug concentration. Furthermore, the weight of the pupal and adult stages was significantly reduced as a result of the impact of ciprofloxacin. Additionally, this study delved into the intricate relationship between ciprofloxacin and the intestinal microflora of *S. peregrina*. Undoubtedly, the presence of ciprofloxacin induced noteworthy alterations in the composition of microorganisms residing within the gastrointestinal tract of *S. peregrina*, thereby inducing a corresponding change in its phenotype. Noteworthy among these intestinal organisms were *Ignatzschineria*, *Providencia*, *Wohlfahrtiimonas*, *Proteus*, *Myroides*, and *Bacteroides*, which seemingly exerted pivotal roles in the growth and development of *S. peregrina*. However, future investigations will provide a more comprehensive elucidation of these intricate matters.

## Figures and Tables

**Figure 1 microorganisms-11-02867-f001:**
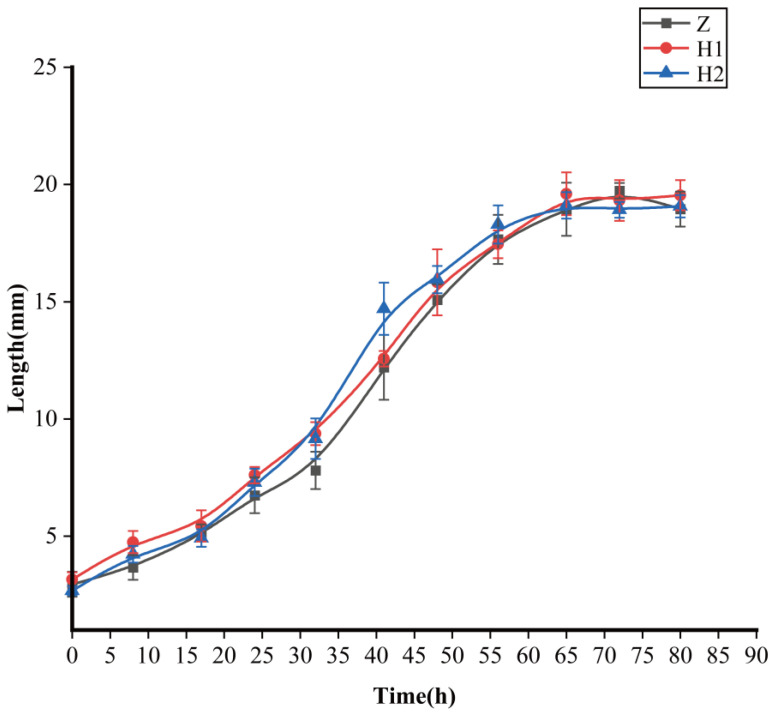
Effects of ciprofloxacin on the growth and development of *S. peregrina*; curves demonstrate the variation of larval body length with developmental time. Z: black curve with rectangles; H1: red curve with dots; H2: blue curve with triangles.

**Figure 2 microorganisms-11-02867-f002:**
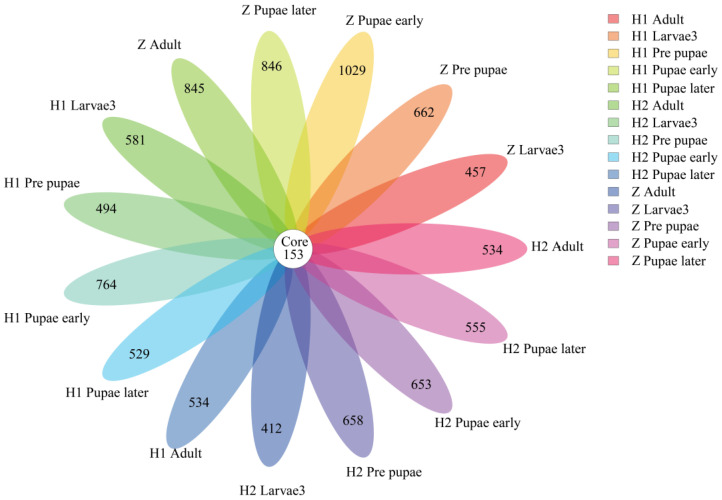
Venn diagram of OTU distribution across life stages of *S. peregrina*. Numbers within compartments indicate the shared OTUs in all samples. OTU: operational taxonomic unit.

**Figure 3 microorganisms-11-02867-f003:**
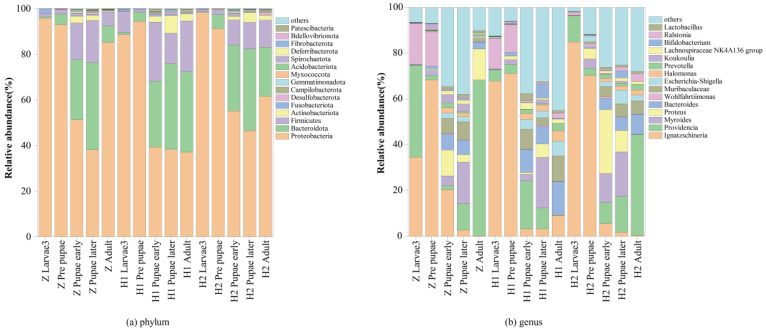
Community structure of the top 15 in different life stages of *S. peregrina*.

**Figure 4 microorganisms-11-02867-f004:**
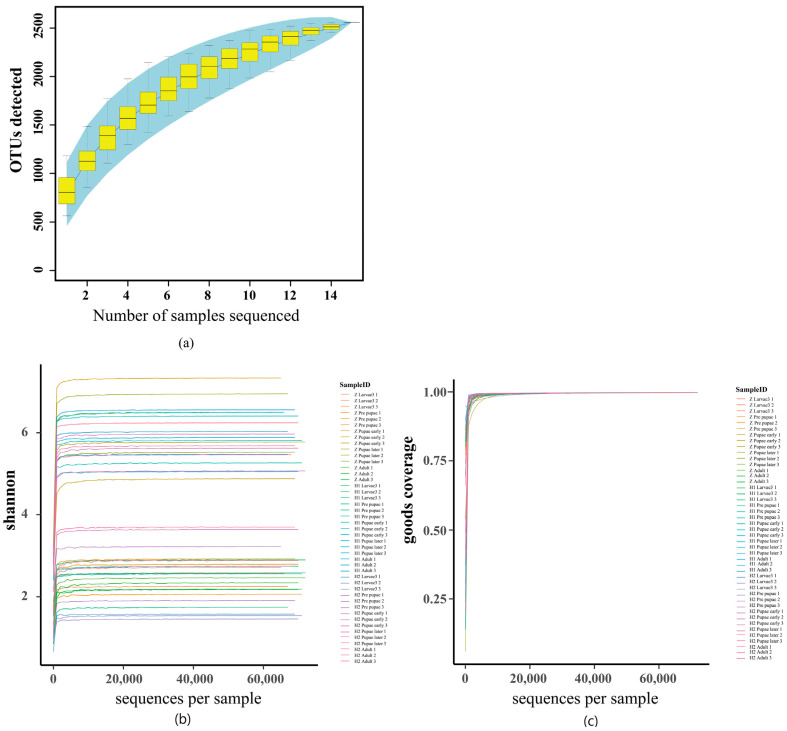
Alpha diversity analysis of microbial composition with three diversity metrics: Specaccum species (**a**), Shannon’s index (**b**), and Good’s coverage index (**c**). The Specaccum species (**a**) accumulation curve tends to be flattened, which suggests that the species in this environment would not increase significantly with a sample size increase, indicating that the sampling is sufficient. The Shannon curve eventually tends to flatten, indicating that the sequencing depth of the sample is large enough (**b**). The Good’s coverage index is close to 1 (**c**), which means that the sequencing depth has covered all species in the sample.

**Figure 5 microorganisms-11-02867-f005:**
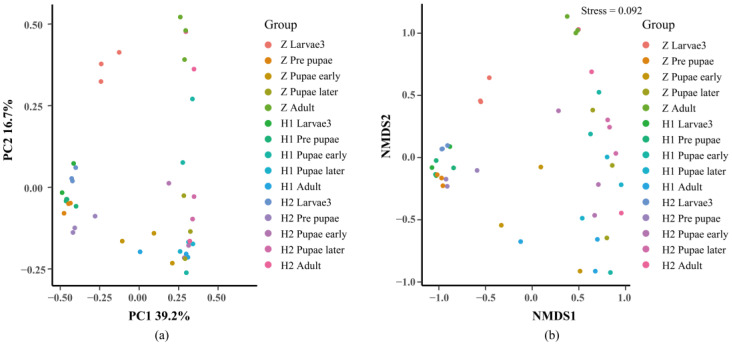
Bata diversity analysis of microbial composition with two diversity metrics: (**a**) PCoA and (**b**) (NMDS). The points in the figure represent samples and different colors represent the different groups. The close distance between points in the same group indicates the strong repeatability of the samples, and further distance between points in different groups indicates greater difference between samples. The contribution of PC1 and PC2 to the difference of the matrix is 39.2% and 16.7%, respectively. The stress value of NMDS was 0.092.

**Figure 6 microorganisms-11-02867-f006:**
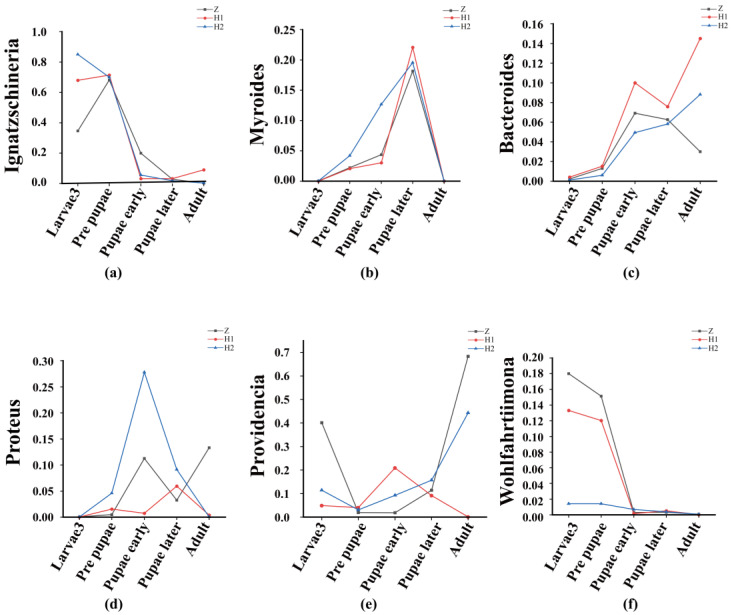
Line charts of the relative abundance of the top 10 at the genus level. (**a**) *Ignatzschineria*, (**b**) *Myroides*, (**c**) *Bacteroides*, (**d**) *Proteus*, (**e**) *Providencia*, (**f**) *Wohlfahrtiimonas*.

**Table 1 microorganisms-11-02867-t001:** One-way analysis of variance. Column averages followed by different letters indicate significant differences between treatments (*p* < 0.05).

	Greatest Length	Pupal Weight	Adult Weight
C	19.74 ± 0.33 a	68.82 ± 4.07 b	41.60 ± 3.69 b
H1	19.60 ± 0.92 a	59.14 ± 1.55 a	38.17 ± 5.02 b
H2	19.11 ± 0.56 a	62.28 ± 2.19 a	22.00 ± 6.75 a
df	2	2	2
F	1.32	15.43	22.88

**Table 2 microorganisms-11-02867-t002:** Adonis (PERMANOVA) analysis result (Bray–Curtis). *** indicates a significant difference.

	Df	Sums of Sqs	MeanSqs	F.Model	R2	*p*-Value (>F)	Signif
Group factor	14	9.115	0.65107	5.6351	0.7245	0.001	***
Residuals	30	3.4662	0.11554		0.2755		
Total	44	12.5812			1		

## Data Availability

The raw data generated in this study can be found in the NCBI Sequence Read Archive under accession PRJNA1028944.

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
