# Peer review of "Effect of Ciprofloxacin on the Composition of Intestinal Microbiota in Sarcophaga peregrina (Diptera: Sarcophagidae)"

_microorganisms, 2023, doi:10.3390/microorganisms11122867_

Round 1

Reviewer 1 Report

Comments and Suggestions for Authors

The document titled "Effect of ciprofloxacin on the composition of the intestinal microbiota in Sarcophaga peregrina contains" valuable information that should be published considering that the insect area and an excellent model to understand the effects of antibiotics on the microbiota of organisms and its effect on its development.

Some observations are described in the pdf document.

Materials and Methods

2.1 Experimental design

Add a section explaining the experimental design. The authors describe it in the first section, three experimental groups are used, one corresponds to the control, and they also use two concentrations of ciprofloxacin. If the control group is only for reference, a two-factor design remains the experimental group (two levels H1, H2), concentration (2 levels, H1 = 0.111μg/g, and H2 =1.33μg/g,).

2.2 Cultivation of S. peregrina

2.6 Statistical analysis
This section should describe how the response variables were analyzed, how statistical differences were determined, how alpha and beta diversity were estimated, etc.

Results

The quality of the figures needs to be improved. The x and y axes should be thickened and the legends should be increased in size, the same with the labels.

Discussion

The discussion in general is well written, but it requires highlighting the effect of Ciprofloxacin on the effect of the microbiome and its effect on the development of organisms.

Reviewer 2 Report

Comments and Suggestions for Authors

I read with great attention the manuscript of H.Tang with colleagues submitted to the journal Microorganisms.

Although I myself am not a microbiologist, but an entomologist interested in studying evolution, this article is interesting to me because the larvae of dipterans living in such nutrient media as meat withstand extremely difficult competition with microorganisms. The work is based on solid data and I find its publication useful, not only for microbiologists, but also for entomologists and dipterologists involved in insect breeding.

At the same time, for a broader general biological and ecological sound of the article, it is necessary in the discussion to at least briefly touch on the more general aspects of the relationship between bacteria and insects, which compete for such highly nutritious substrates as meat.

One of the conclusions of the article reads: “The outcomes of this experiment reveal a relatively higher diversity in the gut microbiome composition of S. peregrina compared to previously reported Diptera species. This could be attributed to the necrophilic nature of these flies, which predisposes them to associate with decaying matter. Furthermore, dietary and environmental factors may influence the diversity of the gut microbiome in these host insects.” On the one hand, this conclusion seems expected, but in fact it is not so obvious. Living in environments rich in microorganisms has led to the appearance in blowflies of a rich arsenal of their own antimicrobial peptides, well studied for example in Calliphora vicina (e.g. doi: 10.1371/journal.pone.0173559) and S. peregrina (e.g. https://doi.org/10.1111/j.1432- 1033.1993.tb18287.x). From the point of view of these data, one would expect a rather lower level of bacterial microflora. It would be interesting to see a more detailed discussion of this aspect of the problem.

Minor corrections:

line 51 Chrysomya Pretoria replace with Chrysomya pretoria

Line 295 “S. peregrina varies” replace with  S. peregrina varies”

Line 297  Parasarcophaga Similis replace with Parasarcophaga similis

Line 298 Pholia Regina replace with Pholia regina

Line 298 Musca domestic replace with Musca domestica

Line 310 M. domestic replace with M. domestica
